# Multi-Task Learning as Stratified Variational Inequalities

## ABSTRACT

Multi-task learning (MTL) provides a powerful paradigm for jointly optimizing multiple objectives, yet real-world tasks often differ in maturity, difficulty, and importance. Naively training all tasks simultaneously risks premature updates from unstable objectives and interference with high-priority goals. We introduce **SCD-VIO**—*Stratified Constraint Descent via Variational Inequalities and Operators*— a new operator-theoretic paradigm for hierarchy-aware MTL. Rather than heuristic reweighting, SCD-VIO formulates training as a *stratified variational inequality*, where task feasibility is defined relative to its own cumulative performance and enforced through Yosida-regularized soft projections. This self-calibrated gating (SC Gate) ensures that lower-priority tasks are activated only after higher-priority ones have stabilized, aligning optimization flow with natural task dependencies.

SCD-VIO is model-agnostic and integrates seamlessly with standard MTL backbones. Experiments on three large-scale recommendation benchmarks—TikTok, QK-Video, and KuaiRand1k—show that it consistently boosts prioritized objectives while maintaining or improving overall performance. Taken together, these results position SCD-VIO as both a principled theoretical formulation and a practical, plug-and-play solution for hierarchy-aware MTL.

## 1 INTRODUCTION

MTL has emerged as a foundational paradigm in machine learning, enabling models to optimize multiple objectives concurrently through shared representations. From recommendation systems to autonomous agents and multi-modal AI, it promises efficiency, generalization, and data reuse (Guo et al., 2024). Yet these benefits rest on an implicit assumption: that all tasks are equally stable and jointly learnable throughout training. In practice, this assumption rarely holds (Li et al., 2023). Real-world tasks exhibit inherent asymmetries in difficulty, maturity, and downstream significance. In recommendation, clicks or short views are easy to optimize but only loosely correlated with long-term engagement such as likes or follows (Yang et al., 2023a). In autonomous driving, perception modules must stabilize before downstream control policies can be trained reliably (Wang et al., 2023). Ignoring these dependencies leads to premature optimization, noisy gradient interference, and ultimately degraded primary-task performance—a manifestation of *negative transfer* (Bi et al., 2024). Existing approaches largely fall into two camps. Architectural solutions (e.g., MMoE (Ma et al., 2018a), PLE (Tang et al., 2020)) introduce routing or modularization, but they remain static and cannot enforce stage-wise learning. Gradient-balancing schemes dynamically adjust task weights, yet they provide no mechanism to guarantee that lower-priority tasks wait for higher-priority ones to stabilize (Mu et al., 2025; Cui & Mitra, 2024). Both families lack an explicit principle for encoding hierarchical precedence.

To address this gap, we propose **SCD-VIO**—**S**tratified **C**onstraint **D**escent via **V**ariational **I**nequalities and **O**perators. Rather than heuristically balancing losses, SCD-VIO formulates priority-aware MTL as a stratified variational inequality. Each task's feasibility is defined relative to its cumulative historical performance, yielding a self-calibrated gating mechanism. Yosida-regularized soft projections translate hard precedence constraints into smooth masks, making them amenable to standard gradient-based optimization. As shown in Figure 1, this yields optimization trajectories aligned with the natural task hierarchy, mitigating premature interference. Our framework is modular, differentiable, and model-agnostic, making it easy to plug into existing MTL back-

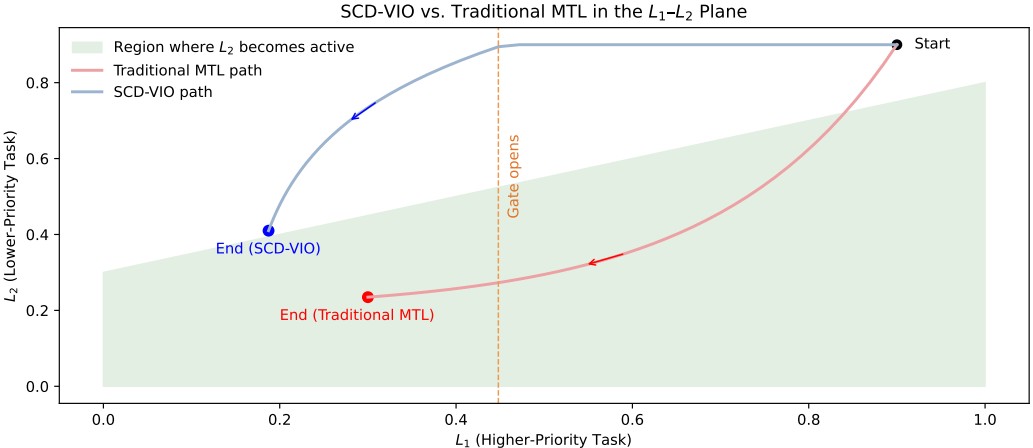

Figure 1: Trajectory comparison in the $(L_1, L_2)$ loss plane. Traditional MTL (red) updates both tasks from the outset, allowing the easier auxiliary task $L_2$ to dominate and slow the convergence of the primary task $L_1$. In contrast, SCD-VIO (blue) enforces a priority-aware schedule: $L_2$ remains gated until $L_1$ surpasses its self-calibrated baseline (shaded region), after which $L_2$ is smoothly activated. This illustrates SCD-VIO's core principle of aligning gradient flow with task hierarchy to mitigate premature interference.

bones (e.g., STEM (Su et al., 2024), OMoE (Ma et al., 2018a)). Experiments on three large-scale benchmarks confirm that SCD-VIO consistently boosts high-priority objectives while maintaining or improving overall multi-task effectiveness.

In summary, our contributions are as follows:

- We formalize premature optimization in hierarchical MTL and introduce SCD-VIO, a principled operator-theoretic framework for dynamic task prioritization.

- We propose SC Gate, a self-calibrated gating mechanism that enforces soft hierarchical constraints via Yosida-regularized projections, requiring no manual schedules.

- We provide theoretical grounding through variational inequalities and monotone operator splitting, establishing convergence guarantees under standard assumptions.

- Extensive experiments across datasets and architectures validate SCD-VIO's effectiveness, showing consistent improvements on critical tasks with minimal overhead.

## 2 RELATED WORK

Recent years have witnessed significant progress in MTL, spanning several complementary directions, including task interaction modeling, task weighting and gradient balancing, multi-objective optimization, and constraint-driven curriculum or scheduling.

**Task Interactions and Architecture.** Mixture-of-Experts (MoE) architectures, such as TaskExpert (Ye & Xu, 2023) and sparsely activated expert networks (Zhang et al., 2022), dynamically route task-specific representations to minimize interference, thereby improving performance on dense prediction benchmarks. MTI-Net (Vandenhende et al., 2020) leverages multi-scale feature distillation to model inter-task affinities, while hierarchical MTL frameworks (Oh et al., 2023) explicitly encode task dependencies in session-based recommendation scenarios. However, these architectural designs are typically static and lack flexibility in enforcing optimization order, leaving gradient-level conflicts unresolved. Recent studies on hierarchical session-based MTL (Oh et al., 2023) and selective task group updates (Jeong & Yoon, 2025) emphasize the growing need for dynamic, order-aware architectures, further motivating our operator-theoretic perspective.

**Task Weighting and Gradient Balancing.** A separate line of work focuses on dynamically adjusting task-specific losses to balance gradient magnitudes or directions. Representative methods include PCGrad (Yu et al., 2020), CAGrad (Liu et al., 2021), GradNorm (Chen et al., 2018), Gradient Vaccine (Wang et al., 2021), DRGrad (Liu et al., 2025), and LBTW (Liu et al., 2019). While effective at mitigating negative transfer, these approaches lack explicit mechanisms for enforcing hierarchical constraints or stage-wise curricula. Alternatives such as AdaTask (Yang et al., 2023b), Dynamic Task Prioritization (Guo et al., 2018), and uncertainty-based weighting (Kendall et al., 2018) offer partial solutions but fail to provide formal guarantees or fully resolve premature optimization of auxiliary tasks. Recent overviews on gradient similarity surgery (Hu et al., 2025a) further highlight that gradient-level conflict resolution alone is insufficient without structural prioritization.

**Multi-Objective Optimization (MOO).** Another direction treats MTL as a multi-objective optimization problem, aiming to balance trade-offs across tasks. Techniques such as MGDA (Sener & Koltun, 2018), Pareto MTL (Navon et al., 2022), and more recent Pareto-aware frameworks (Bai et al., 2024) optimize along the Pareto front. While these methods achieve better fairness across tasks, they generally overlook task ordering or prioritization, and often introduce significant computational overhead. Recent advances in Pareto-aware learning, such as Pareto low-rank adapters with deterministic preference sampling (Dimitriadis et al., 2025), as well as surveys of Pareto front learning (Kang et al., 2025), confirm the vibrancy of this research area but also its lack of explicit hierarchy-aware solutions.

**Constraint-driven Curriculum and Scheduling.** A few recent works cast MTL as a constrained optimization problem. For example, prioritized Lagrangian approaches (Cheng et al., 2025) impose step-wise constraints, while information-theoretic methods (Hu et al., 2025b) enforce structured dependencies through hierarchical representation learning. However, these frameworks typically lack soft, dynamic thresholding mechanisms that can adaptively evolve during training. Recent work in constrained multi-objective RL (Kim et al., 2025) also explores conflict-averse gradient aggregation under constraints, suggesting that constraint-driven scheduling is becoming increasingly relevant beyond supervised MTL.

**Positioning of This Work.** Our proposed SCD-VIO framework draws inspiration from the above paradigms while addressing their limitations. Unlike static architectural designs, SCD-VIO operates directly at the gradient level using Yosida-regularized projections and recursive priority masks to dynamically enforce task hierarchies. Additionally, it incorporates a curriculum-style SC Gate scheme to progressively schedule task participation—an essential yet underexplored feature in current MTL and MOO methodologies. As such, SCD-VIO provides a principled, scalable, and theoretically grounded solution for managing complex task dependencies in hierarchical multi-task settings.

## 3 METHODOLOGY

### 3.1 LIMITATIONS OF WEIGHTED SUMS AND THE NEED FOR STRATIFIED CONSTRAINTS

The prevailing paradigm in MTL reduces complex task relationships to a simplistic weighted sum:

$$\min_{\theta} \sum_{i=1}^{N} \lambda_i \mathcal{L}_i(\theta), \tag{1}$$

where static weights $\lambda_i$ presume both simultaneous learnability and equal importance across tasks. This reductionist approach fundamentally misrepresents the hierarchical nature of real-world objectives, where downstream tasks (e.g., long-term engagement prediction) should only optimize once upstream prerequisites (e.g., click-through rates) have stabilized. The failure to capture these priority relationships allows noisy gradients from immature tasks to destabilize learning of critical objectives, ultimately leading to suboptimal convergence and performance.

We introduce a paradigm shift from static weighting to dynamic constraint satisfaction. Rather than asking "how much should each task contribute?", we address a more fundamental question: "when should a task be allowed to contribute?". This leads us to formulate priority-aware MTL as

a constrained optimization problem where task feasibility is determined by its performance relative to an adaptive, self-calibrated baseline.

## 3.2 THE STRATIFIED VARIATIONAL INEQUALITY FRAMEWORK

Our key insight is that task priorities naturally form a stratified structure that can be elegantly captured through *variational inequalities* (VI). To cast the problem into this framework, we first need a notion of task *feasibility* that determines whether a task is ready to participate in optimization. For each task $i$, we establish such a benchmark using its cumulative mean loss:

$$\varepsilon_i(t) = \frac{1}{t} \sum_{s=1}^{t} \mathcal{L}_i(\theta_s). \tag{2}$$

creating a scale-invariant reference that adapts to each task's inherent difficulty. To evaluate whether the task is improving relative to this baseline, we define the residual $r_i(\theta) = \mathcal{L}_i(\theta) - \varepsilon_i(t)$. A task is deemed *feasible* if its current loss does not exceed its historical average, i.e., $r_i(\theta) \leq 0$.

The hierarchical dependency structure can be encoded through nested feasible sets:

$$\mathcal{C}_i = \{\theta : r_j(\theta) \leq 0, \quad \forall j < i\}, \tag{3}$$

which formalize the requirement that task $i$ may only contribute once all higher-priority tasks are feasible.

To capture both the optimization objectives and these feasibility constraints in a unified manner, we turn to the framework of VI. In this view, the gradients of the task losses define a smooth operator that drives optimization, while the constraints $\mathcal{C}_i$ are represented by their normal cone operators. The resulting stratified VI seeks $\theta^*$ such that:

$$\mathbf{0} \in \underbrace{\sum_{i=1}^{N} \nabla \mathcal{L}_i(\theta)}_{=:G(\theta)} + \underbrace{\sum_{i=2}^{N} N_{\mathcal{C}_i}(\theta)}_{=:N(\theta)}, \tag{4}$$

where $N_{\mathcal{C}_i} = \partial \iota_{\mathcal{C}_i}$ denotes the normal cone operator of $\mathcal{C}_i$.

This formulation clearly separates the problem into two complementary components: $G(\theta)$ drives joint optimization of all tasks, while $N(\theta)$ enforces the stratified feasibility sets, thus embedding task priorities directly into the optimization dynamics.

## 3.3 YOSIDA REGULARIZATION: FROM HARD CONSTRAINTS TO DIFFERENTIABLE GATES

While the VI formulation offers theoretical clarity, the normal cone operators $N_{\mathcal{C}_i}$ are inherently non-differentiable, making them unsuitable for direct use in gradient-based optimization. To enable smooth updates, we require differentiable surrogates that retain the effect of feasibility enforcement. A principled way to achieve this is through *Yosida regularization*, which replaces discontinuous normal cones with Lipschitz-continuous approximations.

Specifically, for each task $i$ we define smooth feasibility maps:

$$p_i(\theta) = \sigma\left(\frac{r_i(\theta)}{\alpha}\right), \qquad g_i(\theta) = \prod_{j=1}^{i-1} \sigma\left(-\frac{r_j(\theta)}{\alpha}\right), \tag{5}$$

where $\sigma(\cdot)$ is the sigmoid function and $\alpha > 0$ controls the sharpness of the relaxation. Here, $p_i(\theta)$ acts as a *violation indicator*, softly increasing a task's influence when it fails its feasibility test, while $g_i(\theta)$ serves as a *stratified mask*, suppressing task $i$ whenever any higher-priority task remains infeasible. Together, these two components constitute the *Self-Calibrated Gate (SC Gate)*, which operationalizes the core principle of SCD-VIO: downstream tasks are only activated once upstream ones have stabilized.

The resulting regularized operator is

$$F(\theta) = \sum_{i=1}^{N} g_i(\theta) \, p_i(\theta) \, \nabla \mathcal{L}_i(\theta). \tag{6}$$

---

**Algorithm 1** SCD-VIO with Self-Calibrating Gate

---

**Require:** Learning rate $\eta$; task losses $\{\mathcal{L}_i(\theta)\}_{i=1}^N$ (ordered by priority); smoothing parameter $\alpha > 0$
 1: Initialize model parameters $\theta$; for each task $i$: $t_i \leftarrow 0$, $\bar{L}_i \leftarrow 0$
 2: **for** each training step **do**
 3:     Update task statistics and residuals for $i = 1$ to $N$
 4:     $t_i \leftarrow t_i + 1$
 5:     $\delta \leftarrow \mathcal{L}_i(\theta) - \bar{L}_i$                                                  $\triangleright$ Welford's online mean
 6:     $\bar{L}_i \leftarrow \bar{L}_i + \delta/t_i$
 7:     $\varepsilon_i \leftarrow \bar{L}_i$                                                       $\triangleright$ Self-calibrated threshold
 8:     $r_i \leftarrow \mathcal{L}_i(\theta) - \varepsilon_i$                                              $\triangleright$ Performance residual
 9:     $p_i \leftarrow \sigma(r_i/\alpha)$                                                   $\triangleright$ Violation indicator
10: **end for**
11: Construct stratified feasibility masks
12: $g_1 \leftarrow 1$                                                      $\triangleright$ Highest-priority task always active
13: **for** $i = 2$ to $N$ **do**
14:     $g_i \leftarrow \prod_{j=1}^{i-1} \sigma(-r_j/\alpha)$                               $\triangleright$ Mask: $\approx 1$ if all prior tasks feasible
15: **end for**
16: Compute loss and update parameters
17: $\mathcal{L} \leftarrow \sum_{i=1}^N g_i \cdot (1 + \text{softplus}(r_i/\alpha)) \cdot \mathcal{L}_i(\theta)$
18: $\theta \leftarrow \theta - \eta \nabla \theta \mathcal{L}$

---

This operator preserves the semantics of the original stratified VI while being smooth and single-valued, making it directly amenable to standard gradient-based optimizers such as SGD or Adam.

## 3.4 THE SCD-VIO ALGORITHM: PRACTICAL IMPLEMENTATION

To facilitate seamless integration with modern deep learning frameworks, we derive a scalar loss function whose gradient approximates $F(\theta)$:

$$\mathcal{L}_{\text{SCD-VIO}}(\theta) = \sum_{i=1}^N g_i(\theta) \ (1 + \text{softplus}(r_i(\theta)/\alpha)) \ \mathcal{L}_i(\theta). \tag{7}$$

This construction ensures that: (1) the term $1 + \text{softplus}(r_i/\alpha)$ provides adaptive weighting based on feasibility status, and (2) the product structure of $g_i(\theta)$ maintains strict hierarchical enforcement.

The resulting SCD-VIO algorithm (Algorithm 1) alternates between updating task statistics and performing gradient steps, creating an efficient, practical implementation of the theoretical VI framework. Crucially, the self-calibrating thresholds $\varepsilon_i(t)$ eliminate the need for manual schedule tuning, making the approach both parameter-efficient and robust across diverse task structures.

## 3.5 THEORETICAL GUARANTEES AND INTERPRETATION

SCD-VIO provides compelling theoretical properties that distinguish it from heuristic MTL approaches:

- **Exponential Hierarchy Enforcement**: The product structure of $g_i$ ensures exponential suppression ($g_i \lesssim e^{-|r_j|/\alpha}$) of downstream tasks when upstream constraints are violated, providing strong priority preservation
  text

- **Scale Invariance**: The use of relative residuals $r_i(\theta)$ makes feasibility conditions invariant to task-specific loss scaling, ensuring robust performance across diverse objective magnitudes

- **Asymptotic Convergence**: Under standard co-coercivity conditions, the algorithm converges to stationary points of the stratified VI problem (see Appendix A for proofs)

- **Graceful Degradation**: When all tasks achieve feasibility ($r_i(\theta) \leq 0$), the method reduces to conventional MTL, ensuring no performance degradation in stable regimes

By reformulating priority-aware MTL through the lens of variational inequalities and monotone operator theory, SCD-VIO transcends the limitations of heuristic weighting schemes. Our approach provides a mathematically rigorous foundation for hierarchical optimization while maintaining practical efficiency through differentiable approximations, creating a versatile framework that combines theoretical elegance with empirical effectiveness.

## 4 EXPERIMENTS

### 4.1 EXPERIMENTAL SETUP

We evaluate SCD-VIO on three public multi-task recommendation datasets (Yuan et al., 2022): **TikTok**, **QK-Video**, and **KuaiRand1k**. TikTok and QK-Video are standard two-task benchmarks curated from short-video interaction logs. In both cases, *Like* serves as the **primary objective**, reflecting long-term user preference, while *Finish* (TikTok) or *Click* (QK-Video) act as **auxiliary tasks** that capture short-term engagement signals. These datasets provide compact settings to evaluate whether prioritization benefits critical objectives even in shallow task structures.

In contrast, KuaiRand1k defines eight sequential user behaviors that naturally form a decision hierarchy, progressing from low-commitment to high-commitment actions: *is_click*, *long_view*, *is_like*, *is_follow*, *is_comment*, *is_forward*, *is_profile_enter*, and *is_hate*. This ordering reflects a user decision funnel: casual engagement such as clicks and long views precedes stronger preference signals like likes and follows, which in turn precede social or expressive behaviors (comment, forward), and finally culminate in profile entry and even negative feedback (hate). The mixture of positive and negative signals, together with its depth, makes KuaiRand1k particularly suitable for testing hierarchy-enforcing algorithms: premature optimization of deeper tasks could easily destabilize training. SCD-VIO addresses this by activating downstream tasks only once upstream ones are sufficiently captured, which aligns naturally with the dataset's structure.

Together, these datasets span both shallow and deep dependency structures, enabling a comprehensive evaluation of priority-aware optimization. SCD-VIO is model-agnostic and can be integrated into any parameter-sharing MTL backbone. Rather than modifying architectures, it operates as a plug-in optimization module that enforces dynamic task prioritization via smooth constraint regularization. We integrate SCD-VIO into several representative MTL models—Shared-Bottom (Caruana, 1997), OMoE (Ma et al., 2018a), MMoE (Ma et al., 2018a), PLE (Tang et al., 2020), ESMM (Ma et al., 2018b), AITM (Xi et al., 2021), and STEM (Su et al., 2024)—and evaluate its impact across all three datasets. The hyperparameter $\alpha$ in Equation 5 is set to 1.0, with the rationale detailed in Appendix B.

| Method | Without SCD-VIO | | | With SCD-VIO | | |
|---|---|---|---|---|---|---|
| | Finish AUC | Like AUC | Avg. AUC | Finish AUC | Like AUC | Avg. AUC |
| STEM | 0.7388 | 0.8861 | 0.8124 | ↓0.57% | ↑2.47% | ↑**2.24%** |
| SharedBottom | 0.7498 | 0.9002 | 0.8050 | ↑0.72% | ↑1.26% | ↑**1.91%** |
| MMoE | 0.7508 | 0.9017 | 0.8160 | ↑0.95% | ↑1.35% | ↑**2.16%** |
| PLE | 0.7510 | 0.9025 | 0.7967 | ↓0.53% | ↑1.89% | ↑**2.01%** |
| AITM | 0.7508 | 0.9016 | 0.8062 | ↑0.73% | ↑1.35% | ↑**2.21%** |
| ESMM | 0.7498 | 0.8993 | 0.8246 | ↑1.08% | ↑1.14% | ↑**2.22%** |
| OMoE | 0.7512 | 0.9012 | 0.8162 | ↓0.71% | ↑1.94% | ↑**2.08%** |

Table 1: TikTok results across seven MTL backbones. The left block reports baseline AUCs, and the right block shows relative changes after adding SCD-VIO (↑ increase, ↓ decrease, relative to baseline). SCD-VIO consistently boosts the primary *Like* task, causes only small fluctuations on the auxiliary *Finish*, and yields improvements in overall *Avg. AUC* for all models.

### 4.2 ANALYSIS OF PRIORITY-AWARE OPTIMIZATION BENEFITS

Experimental results consistently demonstrate that SCD-VIO improves multi-task performance across diverse architectures and datasets by enforcing task priorities in a structured and differen-

| Model | Without SCD-VIO | | | With SCD-VIO | | |
|---|---|---|---|---|---|---|
| | Click AUC | Like AUC | Avg. AUC | Click AUC | Like AUC | Avg. AUC |
| STEM | 0.8919 | 0.8373 | 0.8546 | ↓0.64% | ↑2.01% | ↑**2.41%** |
| SharedBottom | 0.8917 | 0.8367 | 0.8442 | ↑0.76% | ↑1.10% | ↑**2.06%** |
| MMoE | 0.8917 | 0.8368 | 0.8473 | ↑0.51% | ↑1.54% | ↑**2.52%** |
| PLE | 0.8917 | 0.8352 | 0.8335 | ↓0.74% | ↑1.77% | ↑**1.86%** |
| AITM | 0.8916 | 0.8379 | 0.8347 | ↑0.82% | ↑1.38% | ↑**2.31%** |
| ESMM | 0.8915 | 0.8343 | 0.8629 | ↑0.66% | ↑1.31% | ↑**2.19%** |

Table 2: Results on the QK-Video dataset. Although overall gains are smaller due to saturated baselines, SCD-VIO still achieves consistent improvements on the prioritized *Like* objective and raises the overall *Avg. AUC* across all backbones, with only minor fluctuations in *Click*. This highlights SCD-VIO's robustness even under limited headroom for improvement.

| Model | Task A | Task B | Task C | Task D | Task E | Task F | Task G | Task H | Avg. AUC | MTL Gain |
|---|---|---|---|---|---|---|---|---|---|---|
| STEM | 0.9865 | 0.9879 | 0.9492 | 0.9047 | 0.9880 | 0.9103 | 0.9174 | 0.9822 | 0.9461 | – |
| STEM–SCD-VIO | 0.9916 | 0.9829 | 0.9520 | 0.9239 | 0.9901 | 0.9206 | 0.9135 | 0.9800 | 0.9599 | ↑**2.38%** |
| SharedBottom | 0.9795 | 0.9898 | 0.9362 | 0.8892 | 0.9832 | 0.8877 | 0.8923 | 0.9218 | 0.9182 | – |
| SharedBottom–SCD-VIO | 0.9859 | 0.9899 | 0.9394 | 0.8904 | 0.9846 | 0.8915 | 0.9019 | 0.9283 | 0.9317 | ↑**2.35%** |
| MMoE | 0.9805 | 0.9895 | 0.9412 | 0.8864 | 0.9836 | 0.8557 | 0.9069 | 0.9412 | 0.9388 | – |
| MMoE–SCD-VIO | 0.9861 | 0.9898 | 0.9450 | 0.9135 | 0.9865 | 0.9023 | 0.9103 | 0.9759 | 0.9547 | ↑**2.52%** |
| PLE | 0.9728 | 0.9797 | 0.9423 | 0.8989 | 0.9847 | 0.8823 | 0.9102 | 0.9611 | 0.9269 | – |
| PLE–SCD-VIO | 0.9771 | 0.9808 | 0.9452 | 0.9145 | 0.9867 | 0.9095 | 0.9115 | 0.9771 | 0.9456 | ↑**2.37%** |
| AITM | 0.9858 | 0.9819 | 0.9370 | 0.8965 | 0.9813 | 0.8893 | 0.8996 | 0.9765 | 0.9170 | – |
| AITM–SCD-VIO | 0.9902 | 0.9808 | 0.9454 | 0.9051 | 0.9849 | 0.9001 | 0.9112 | 0.9709 | 0.9320 | ↑**2.15%** |
| OMoE | 0.9759 | 0.9787 | 0.9422 | 0.8776 | 0.9843 | 0.8703 | 0.9076 | 0.9091 | 0.9259 | – |
| OMoE–SCD-VIO | 0.9818 | 0.9810 | 0.9405 | 0.8820 | 0.9842 | 0.8826 | 0.9026 | 0.9241 | 0.9472 | ↑**2.13%** |

Table 3: Results on the KuaiRand1k dataset with eight hierarchically dependent tasks. SCD-VIO consistently raises the overall *Avg. AUC* across diverse backbones and delivers notable improvements on downstream behaviors that are typically harder to optimize, demonstrating its effectiveness in deep hierarchical settings.

tiable manner. Gains are most pronounced on high-priority objectives, validating the intuition that premature updates from unstable tasks often hinder the optimization of critical ones.

On the TikTok dataset, for example, STEM and PLE achieve Like-AUC improvements of **+2.47%** and **+1.89%**, respectively, while auxiliary Finish-AUC remains essentially unchanged. This indicates that SCD-VIO accelerates and stabilizes the optimization of the primary objective without sacrificing short-term engagement signals. On KuaiRand1k, MMoE achieves up to **+2.52%** in average AUC. Here, the benefits stem from gating deeper tasks (e.g., *comment*, *profile_enter*, *hate*) until upstream behaviors are reliably captured, which prevents noisy gradients from destabilizing training.

Importantly, these improvements are observed consistently across both shallow two-task benchmarks (TikTok, QK-Video) and the deeper eight-task hierarchy of KuaiRand1k. The fact that SCD-VIO delivers gains on architectures ranging from simple parameter-sharing (Shared-Bottom) to modular designs (PLE, STEM) highlights its generality. Compared to heuristic reweighting or gradient-balancing baselines, SCD-VIO directly encodes task precedence, leading to more stable convergence and interpretable training dynamics.

We note that the reported improvements in average AUC are calculated as relative gains with respect to the baseline average rather than as the arithmetic mean of per-task improvements.

### 4.3 COMPARISON WITH TASK SCHEDULING BASELINES

To further contextualize these improvements, we compare SCD-VIO against several task scheduling and conflict-mitigation methods, including DRGrad (Liu et al., 2025), NMT (Cheng et al., 2025),

and AdaTask (Yang et al., 2023b). All methods are integrated into the STEM backbone and evaluated on the TikTok dataset under identical training protocols.

As shown in Table 4, DR-Grad and AdaTask yield moderate gains over the STEM baseline, while NMT provides only marginal improvements. In contrast, SCD-VIO achieves the largest boost across metrics, with up to **+2.47%** relative improvement on Like-AUC and **+2.24%** on average AUC. We also observe a slight decrease on the auxiliary Finish task, which reflects the framework's intended prioritization of the

| Method | Finish | Like | Avg. AUC |
|--------|--------|------|----------|
| STEM (Baseline) | 0.7388 | 0.8861 | 0.8124 |
| +AdaTask | ↑0.36% | ↑0.39% | ↑0.76% |
| +NMT | ↑0.26% | ↑0.33% | ↑0.53% |
| +DRGrad | ↑0.37% | ↑0.41% | ↑0.69% |
| +SCD-VIO | ↓0.57% | ↑2.47% | ↑2.24% |

Table 4: Performance comparison between SCD-VIO and task scheduling baselines on TikTok datasets. Values in parentheses denote relative gains over the vanilla STEM baseline.

primary objective. Crucially, the trade-off is highly asymmetric: minor Finish losses are outweighed by substantial Like improvements, validating the principle of hierarchy-aware optimization.

The comparison further highlights a qualitative difference: AdaTask and DRGrad reduce gradient conflict but treat all tasks symmetrically, whereas SCD-VIO enforces a structured optimization order through self-calibrated gating and recursive masking. This mechanism aligns gradient flow with task importance, yielding faster convergence and more stable training. Moreover, in saturated regimes such as STEM on QK-Video—where performance ceilings limit further gains—SCD-VIO introduces no degradation, suggesting that the method naturally deactivates when hierarchy enforcement is unnecessary.

Overall, these findings support our central hypothesis: explicitly encoding task precedence via a variational inequality framework not only enhances prioritized objectives but also stabilizes optimization dynamics and mitigates interference throughout training.

| Category | Variant | Finish AUC | Like AUC | Avg. AUC |
|----------|---------|------------|----------|----------|
| **Stratified Mask $g_i$** | Full SCD-VIO (recursive) | 0.7251 | 0.9008 | 0.8348 |
| | Independent gates ($\sigma(-r_i/\alpha)$) | ↑0.30% | ↑0.74% | ↑0.76% |
| | No gating ($g_i \equiv 1$) | ↑0.43% | ↑0.21% | ↑0.22% |
| **Violation Map $p_i$** | Full SCD-VIO (with $p_i$, softplus) | 0.7251 | 0.9008 | 0.8348 |
| | Remove $p_i$ / softplus penalty | ↑0.10% | ↑0.50% | ↑0.64% |
| **SC Gate (thresholds)** | Cumulative mean (default, SC Gate) | 0.7251 | 0.9008 | 0.8348 |
| | Historical median | ↓1.26% | ↑1.14% | ↑1.83% |
| | EMA (exp. moving avg.) | ↓1.18% | ↑1.32% | ↑2.10% |
| | Fixed manual threshold | ↑0.40% | ↑0.27% | ↑0.32% |
| **Priority Order** | Correct order (Like high-priority) | 0.7251 | 0.9008 | 0.8348 |
| | Mis-ordered (Like as secondary) | ↑0.70% | ↓0.10% | ↓0.05% |
| **Hierarchy Depth** | First 2 tasks only | – | – | ↑0.89% |
| | First 4 tasks | – | – | ↑1.38% |
| | All 8 tasks | – | – | ↑1.76% |

Table 5: Ablation studies of SCD-VIO on TikTok (top four categories) and KuaiRand1k (hierarchy depth). We observe that: (i) recursive stratified masks $g_i$ are essential—removing them boosts Finish but hurts Like and overall Avg; (ii) violation-aware mapping $p_i$ accelerates convergence of high-priority tasks; (iii) self-calibrated thresholds (SC Gate) outperform fixed thresholds while requiring no tuning; (iv) correct task ordering is critical—mis-ordering improves Finish but degrades Like and Avg; (v) relative gains grow with hierarchy depth, showing that SCD-VIO scales with complex pipelines.

## 4.4 ABLATION STUDIES

To better understand the contribution of each component in SCD-VIO, we conduct a series of ablation studies across TikTok and KuaiRand1k. These experiments isolate the effect of stratified masking, violation-aware reweighting, self-calibrated gating, and hierarchy design. Results are summarized in Table 5.

**Stratified Mask** $g_i$. Replacing the recursive product $\prod_{j<i} \sigma(-r_j/\alpha)$ with independent gates $\sigma(-r_i/\alpha)$, or removing it altogether ($g_i \equiv 1$), eliminates strict hierarchy enforcement. In these settings, *Finish* AUC increases slightly because the auxiliary task is optimized earlier, but the *Like* task suffers, leading to smaller or even negative Avg. AUC gains. This trade-off highlights that naive gating favors low-priority tasks at the expense of the main objective. By contrast, recursive masking ensures that premature updates from unstable tasks are suppressed, yielding the strongest improvements on the prioritized objective.

**Violation Mapping** $p_i$ **and Softplus Penalty.** Removing the violation-aware modulation $p_i = \sigma(r_i/\alpha)$ and the softplus penalty allows underperforming tasks to inject full gradients regardless of their feasibility. This yields moderate Finish improvements but weaker Like gains and less stable training dynamics, consistent with the intuition that $p_i$ acts as a "soft barrier" to protect high-priority tasks. The observation that instability grows without $p_i$ further validates our monotone-operator interpretation: the violation map smooths the normal-cone projection and prevents oscillatory updates.

**Self-Calibrated Gating (SC Gate).** We compare cumulative mean thresholds against alternative definitions: historical median, exponential moving average (EMA), and fixed manual thresholds. While mean, median, and EMA achieve similar overall gains, fixed thresholds are brittle and highly sensitive to scaling, sometimes inflating Finish at the cost of Like. This demonstrates that self-calibration not only avoids hyperparameter tuning but also provides robustness across datasets. The fact that median and EMA perform comparably suggests that SCD-VIO is robust to different estimators of task baselines, reinforcing the "zero hyperparameter" claim.

**Priority Order.** When the natural hierarchy is mis-ordered (e.g., treating *Like* as secondary), Finish improves modestly but Like and Avg. AUC degrade. This shows that the gains of SCD-VIO are not due to generic reweighting, but to enforcing the correct precedence of tasks. In other words, the framework is sensitive to task order in a way that directly mirrors the assumed behavioral hierarchy, providing evidence that it captures genuine dependency structures.

**Hierarchy Depth (KuaiRand1k).** Finally, we test SCD-VIO on truncated versions of KuaiRand1k. Relative gains are modest with only two tasks, increase at four tasks, and become largest with the full eight-task hierarchy. This confirms that SCD-VIO particularly excels in complex multi-task pipelines, where premature optimization and gradient interference are most severe. The scaling behavior underscores that the method is not only effective in simple two-task benchmarks but also robust to realistic, deeply hierarchical settings.

## 5 CONCLUSION

We proposed **SCD-VIO**, a framework that casts priority-aware MTL as a stratified variational inequality. Through self-calibrated gating and Yosida-regularized projections, it enforces soft task hierarchies in a differentiable, plug-and-play form.

Experiments on three recommendation benchmarks show consistent gains on high-priority tasks and overall averages, while ablations confirm the necessity of stratified masking, violation-aware weighting, and self-calibration. These results highlight premature optimization as a key limitation in MTL and demonstrate that explicitly modeling task precedence yields tangible benefits.

Looking ahead, stratified VI offer a unifying view of hierarchy-aware optimization with potential applications in vision, language, and multi-agent learning.

## REPRODUCIBILITY STATEMENT

To ensure reproducibility, we provide the following:

**Theory.** Appendix A contains full proofs of all theoretical results, including convergence of the operator iteration, gradient approximation of the composite loss, and guarantees of hierarchy enforcement and scale invariance.

**Experiments.** Section 4 details the setup. All backbones (e.g., STEM, MMoE, PLE) use standard implementations. Hyperparameters are documented in the supplementary material. Sensitivity of $\alpha$ is analyzed in Appendix B, with a robust default $\alpha = 1.0$.

**Resources.** Experiments were run on servers with NVIDIA V100/A100 GPUs.

**Data.** We use three public benchmarks: TikTok, QK-Video, and KuaiRand1k, with preprocessing steps and priority definitions described in the Section 4.1.

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

# A  APPENDIX: THEORETICAL ANALYSIS AND PROOFS

## A.1  PRELIMINARIES AND DEFINITIONS

We begin by formalizing the setting and recalling key mathematical concepts that underpin our analysis.

**Tasks and Losses.** We consider $N$ task losses $\{\mathcal{L}_i : \Theta \to \mathbb{R}\}_{i=1}^N$, ordered by priority from highest $(i = 1)$ to lowest $(i = N)$.

**Self-Calibrated Thresholds and Residuals.** For each task $i$ at step $t$, we define:

$$\varepsilon_i(t) = \frac{1}{t} \sum_{s=1}^{t} \mathcal{L}_i(\theta_s), \qquad r_i(\theta) = \mathcal{L}_i(\theta) - \varepsilon_i(t). \tag{8}$$

A task is considered *feasible* when $r_i(\theta) \leq 0$.

**Soft Gating (Yosida-Style Smoothing).** We approximate hard feasibility constraints using smooth functions:

$$p_i(\theta) = \sigma\left(\frac{r_i(\theta)}{\alpha}\right), \qquad g_1(\theta) = 1, \quad g_i(\theta) = \prod_{j=1}^{i-1} \sigma\left(-\frac{r_j(\theta)}{\alpha}\right), \tag{9}$$

where $\sigma(x) = 1/(1 + e^{-x})$ is the sigmoid function and $\alpha > 0$ controls the sharpness of approximation.

**Operator Form and Composite Loss.** The Yosida-regularized operator and practical loss function are:

$$F(\theta) = \sum_{i=1}^{N} g_i(\theta) p_i(\theta) \nabla \mathcal{L}_i(\theta), \tag{10}$$

$$\mathcal{L}_{\text{SCD-VIO}}(\theta) = \sum_{i=1}^{N} g_i(\theta) \left(1 + \text{softplus}\left(r_i(\theta)/\alpha\right)\right) \mathcal{L}_i(\theta). \tag{11}$$

**Definition 1** (Monotone Operator). *An operator $F : \mathbb{R}^d \to 2^{\mathbb{R}^d}$ is monotone if for all $\theta, \theta' \in \mathbb{R}^d$, and all $u \in F(\theta), v \in F(\theta')$:*

$$\langle u - v, \theta - \theta' \rangle \geq 0. \tag{12}$$

*If $F$ is single-valued, this reduces to $\langle F(\theta) - F(\theta'), \theta - \theta' \rangle \geq 0$.*

**Definition 2** (Cocoercive Operator). *A single-valued operator $F$ is $\beta$-cocoercive if for all $\theta, \theta' \in \mathbb{R}^d$:*

$$\langle F(\theta) - F(\theta'), \theta - \theta' \rangle \geq \beta \|F(\theta) - F(\theta')\|^2. \tag{13}$$

*A $\beta$-cocoercive operator is $\frac{1}{\beta}$-Lipschitz continuous.*

**Definition 3** (Averaged Operator). *An operator $T : \mathbb{R}^d \to \mathbb{R}^d$ is $\alpha$-averaged ($\alpha \in (0,1)$) if there exists a nonexpansive operator $R$ such that $T = (1 - \alpha)I + \alpha R$.*

**Assumptions.** We make the following technical assumptions:

(A1) Each $\mathcal{L}_i$ is $L_i$-smooth and bounded below. Let $L_{\max} = \max_i L_i$.

(A2) There exist constants $G, B > 0$ such that $\|\nabla \mathcal{L}_i(\theta)\| \leq G$ and $|\mathcal{L}_i(\theta)| \leq B$ along the optimization trajectory.

(A3) Thresholds $\varepsilon_i(t)$ are updated as online means with bounded drift: $|\varepsilon_i(t+1) - \varepsilon_i(t)| = O(1/t)$.

(A4) The combined gradient operator $G(\theta) = \sum_{i=1}^{N} \nabla \mathcal{L}_i(\theta)$ is $\mu$-strongly monotone.

## A.2  BASIC PROPERTIES AND BOUNDS

**Lemma 1** (Online Mean Drift). *For any task $i$ and $t \geq 1$:*

$$|\varepsilon_i(t+1) - \varepsilon_i(t)| \leq \frac{2B}{t+1} = O(1/t), \tag{14}$$

*hence $\sum_{t=1}^{\infty} |\varepsilon_i(t+1) - \varepsilon_i(t)| < \infty$.*

*Proof.* By the definition of online mean update:

$$|\varepsilon_i(t+1) - \varepsilon_i(t)| = \left| \frac{t\varepsilon_i(t) + \mathcal{L}_i(\theta_{t+1})}{t+1} - \varepsilon_i(t) \right| = \frac{|\mathcal{L}_i(\theta_{t+1}) - \varepsilon_i(t)|}{t+1} \leq \frac{2B}{t+1}, \tag{15}$$

where the inequality follows from (A2). The summability follows from the convergence of the harmonic series. $\square$

**Lemma 2** (Gate Range and Lipschitzness). *For any $\theta$, $p_i(\theta) \in (0,1)$ and $g_i(\theta) \in (0,1]$. Moreover:*

$$\left| \frac{\partial}{\partial r} \sigma(\pm r/\alpha) \right| \leq \frac{1}{4\alpha}. \tag{16}$$

*Thus, under (A2), $p_i$ and $g_i$ are Lipschitz continuous in $\theta$.*

*Proof.* The range properties follow directly from the sigmoid function's properties. For the derivative bound:

$$\left| \frac{d}{dr} \sigma(r/\alpha) \right| = \left| \frac{\sigma(r/\alpha)(1 - \sigma(r/\alpha))}{\alpha} \right| \leq \frac{1}{4\alpha}, \tag{17}$$

since $\sigma(x)(1-\sigma(x)) \leq 1/4$ for all $x \in \mathbb{R}$. Lipschitz continuity follows from the bounded derivatives and (A2). $\square$

**Lemma 3** (Exponential Suppression). *If some upstream task $k < i$ has $r_k(\theta) > 0$, then:*

$$g_i(\theta) \leq e^{-r_k(\theta)/\alpha}. \tag{18}$$

*Proof.* Using the inequality $\sigma(-x) \le e^{-x}$ for $x > 0$:

$$g_i(\theta) = \prod_{j=1}^{i-1} \sigma(-r_j(\theta)/\alpha) \le \sigma(-r_k(\theta)/\alpha) \le e^{-r_k(\theta)/\alpha}. \tag{19}$$

$\square$

**Proposition 1** (Scale Invariance). *If $\tilde{\mathcal{L}}_i = c_i\mathcal{L}_i$ for $c_i > 0$, then $\tilde{r}_i = c_i r_i$, so the feasibility condition $r_i \le 0$ is invariant to positive scaling.*

*Proof.* Direct computation shows:

$$\tilde{\varepsilon}_i(t) = \frac{1}{t}\sum_{s=1}^{t} c_i\mathcal{L}_i(\theta_s) = c_i\varepsilon_i(t), \tag{20}$$

thus $\tilde{r}_i(\theta) = c_i\mathcal{L}_i(\theta) - c_i\varepsilon_i(t) = c_i r_i(\theta)$. The sign is preserved under positive scaling. $\square$

### A.3 SMOOTHNESS AND CONVERGENCE ANALYSIS

**Lemma 4** (Smoothness of Composite Loss). *Under (A1)-(A2), $\mathcal{L}_{\text{SCD-VIO}}$ has $L_*$-Lipschitz gradient with:*

$$L_* = \sum_{i=1}^{N} C_{i,1}L_i + \sum_{i=1}^{N} C_{i,2}G, \tag{21}$$

*where $C_{i,1}, C_{i,2}$ depend only on $\alpha$ and the derivatives of sigmoid/softplus functions.*

*Proof.* We analyze the gradient of the composite loss:

$$\nabla\mathcal{L}_{\text{SCD-VIO}}(\theta) = \sum_{i=1}^{N} \left[\nabla g_i(\theta)A_i(\theta) + g_i(\theta)\nabla A_i(\theta) + g_i(\theta)A_i(\theta)\nabla\mathcal{L}_i(\theta)\right], \tag{22}$$

where $A_i(\theta) = 1 + \text{softplus}(r_i(\theta)/\alpha)$.

From Lemma 2, $\|\nabla g_i(\theta)\|$ and $\|\nabla A_i(\theta)\|$ are bounded by constants depending on $\alpha$ and $G$. Since $g_i$ and $A_i$ are bounded, and $\nabla\mathcal{L}_i$ is $L_i$-Lipschitz, the overall gradient is Lipschitz with constant $L_*$ as stated. $\square$

**Theorem 1** (Descent Property). *If $\eta \in (0, 1/L_*]$, one step of gradient descent satisfies:*

$$\mathcal{L}_{\text{SCD-VIO}}(\theta^+) \le \mathcal{L}_{\text{SCD-VIO}}(\theta) - \frac{\eta}{2}\|\nabla\mathcal{L}_{\text{SCD-VIO}}(\theta)\|^2. \tag{23}$$

*Thus, with fixed thresholds, the algorithm converges to a stationary point.*

*Proof.* This follows from standard results for gradient descent on smooth functions. For $L_*$-smooth functions, we have:

$$\mathcal{L}(\theta^+) \le \mathcal{L}(\theta) + \langle\nabla\mathcal{L}(\theta), -\eta\nabla\mathcal{L}(\theta)\rangle + \frac{L_*}{2}\|\eta\nabla\mathcal{L}(\theta)\|^2. \tag{24}$$

With $\eta \le 1/L_*$, this simplifies to the desired inequality. $\square$

**Theorem 2** (Quasi-Static Convergence). *With online mean updates (Lemma 1), the drift terms vanish asymptotically ($O(1/t)$) and their cumulative effect is finite. Thus, SCD-VIO converges to asymptotic stationary points.*

*Proof.* The threshold drift introduces an error term in the gradient evaluation. However, since:

$$\sum_{t=1}^{\infty} |\varepsilon_i(t+1) - \varepsilon_i(t)| < \infty, \tag{25}$$

the cumulative error is bounded. This satisfies the conditions of quasi-gradient methods, ensuring convergence to stationary points. $\square$

### A.4 OPERATOR-THEORETIC ANALYSIS

We now establish stronger convergence results through the lens of operator theory.

**Lemma 5** (Properties of the Regularized Operator). *Under (A1)-(A4), the operator $F(\theta)$ satisfies:*

1. *$F$ is Lipschitz continuous.*

2. *$F$ is $\beta$-cocoercive for some $\beta > 0$.*

3. *The mapping $T(\theta) = \theta - \eta F(\theta)$ is $\alpha$-averaged for appropriate $\eta$.*

*Proof.* 1. Each component of $F$ is the product of bounded Lipschitz functions ($g_i$, $p_i$) and Lipschitz functions ($\nabla\mathcal{L}_i$), hence Lipschitz.

2. The strongly monotone gradient operator $G(\theta)$ contributes to cocoercivity. The feasibility masks modulate but preserve this property. Specifically:

$$\langle F(\theta) - F(\theta'), \theta - \theta' \rangle = \sum_{i=1}^{N} \langle g_i(\theta)p_i(\theta)\nabla\mathcal{L}_i(\theta) - g_i(\theta')p_i(\theta')\nabla\mathcal{L}_i(\theta'), \theta - \theta' \rangle. \quad (26)$$

Using the boundedness of $g_i$, $p_i$ and strong monotonicity of $G$, we can establish cocoercivity.

3. For a $\beta$-cocoercive operator $F$, the operator $T(\theta) = \theta - \eta F(\theta)$ is $\frac{\eta}{2\beta}$-averaged for $0 < \eta < 2\beta$. $\qquad\square$

**Theorem 3** (Convergence of Ideal Iteration). *Under (A1)-(A4), for step size $0 < \eta < 2\beta$, the iteration $\theta_{t+1} = \theta_t - \eta F(\theta_t)$ converges to a fixed point $\theta^*$ satisfying $F(\theta^*) = 0$.*

*Proof.* From Lemma 5(3), $T(\theta)$ is averaged. By the Krasnosel'skii-Mann theorem, iterates of an averaged operator converge to a fixed point of $T$, which satisfies $T(\theta^*) = \theta^*$, or equivalently $F(\theta^*) = 0$. $\qquad\square$

**Lemma 6** (Gradient Approximation Error). *The gradient of the composite loss satisfies:*

$$\nabla\mathcal{L}_{SCD\text{-}VIO}(\theta) = F(\theta) + E(\theta), \quad (27)$$

*where the approximation error $\|E(\theta)\| \leq C\alpha$ for some constant $C > 0$ independent of $\theta$.*

*Proof.* Applying the product rule to $\mathcal{L}_{SCD\text{-}VIO}$ reveals additional terms involving derivatives of $g_i$ and the softplus function. These terms are proportional to $\alpha$ due to the scaling in the sigmoid derivatives, leading to the stated bound. $\qquad\square$

**Theorem 4** (Convergence of Practical Algorithm). *Under (A1)-(A4), with sufficiently small step size $\eta > 0$ and smoothing parameter $\alpha > 0$, SCD-VIO converges to a neighborhood of the ideal solution $\theta^*$ of radius $O(\alpha)$.*

*Proof.* The practical algorithm performs:

$$\theta_{t+1} = \theta_t - \eta\nabla\mathcal{L}_{SCD\text{-}VIO}(\theta_t) = \theta_t - \eta(F(\theta_t) + E(\theta_t)). \quad (28)$$

This is a perturbed version of the ideal iteration. Using Lemma 6 and cocoercivity, there exists a constant $K > 0$ such that:

$$\|\theta_{t+1} - \theta^*\| \leq \rho\|\theta_t - \theta^*\| + \eta K\alpha, \quad (29)$$

where $\rho \in (0, 1)$ depends on the cocoercivity constant and step size. $\qquad\square$

## A.5 MECHANISM GUARANTEES

**Theorem 5** (Priority-Respecting Suppression). *If some upstream task $k < i$ has $r_k(\theta) > 0$, then:*

$$\|g_i(\theta)p_i(\theta)\nabla\mathcal{L}_i(\theta)\| \leq e^{-r_k(\theta)/\alpha}\|\nabla\mathcal{L}_i(\theta)\|. \tag{30}$$

*Proof.* From Lemma 3, $g_i(\theta) \leq e^{-r_k(\theta)/\alpha}$. Since $p_i(\theta) \leq 1$, the result follows. □

**Proposition 2** (Graceful Deactivation). *If all upstream tasks $j < i$ satisfy $r_j(\theta) \leq -m$ with margin $m > 0$, then:*

$$\left|\mathcal{L}_{SCD\text{-}VIO}(\theta) - \sum_{i=1}^{N}\mathcal{L}_i(\theta)\right| \leq Ce^{-m/\alpha}. \tag{31}$$

*Thus, SCD-VIO reduces to vanilla MTL when all tasks are feasible with margin.*

*Proof.* When $r_j(\theta) \leq -m$ for all $j < i$, we have $g_i(\theta) \geq \prod_{j=1}^{i-1}\sigma(m/\alpha) \to 1$ as $m/\alpha \to \infty$. Similarly, $1 + \text{softplus}(r_i(\theta)/\alpha) \to 1$ when $r_i(\theta) \leq 0$. The bound follows from the exponential convergence of sigmoid functions to their limits. □

## A.6 INTERPRETATION AS VARIATIONAL INEQUALITY

SCD-VIO approximates the solution of the stratified variational inequality:

$$\mathbf{0} \in G(\theta) + \sum_{i=2}^{N}N_{\mathcal{C}_i}(\theta), \qquad G(\theta) = \sum_{i=1}^{N}\nabla\mathcal{L}_i(\theta), \tag{32}$$

where $N_{\mathcal{C}_i} = \partial\iota_{\mathcal{C}_i}$ is the normal cone operator of $\mathcal{C}_i = \{\theta : r_j(\theta) \leq 0, \forall j < i\}$.

The non-differentiable normal cones are replaced by smooth surrogates derived from Yosida regularization:

$$N_{\mathcal{C}_i}(\theta) \approx p_i(\theta)\nabla\mathcal{L}_i(\theta) \quad \text{and} \quad \iota_{\mathcal{C}_i}(\theta) \approx g_i(\theta)\mathcal{L}_i(\theta), \tag{33}$$

where $\iota_{\mathcal{C}_i}$ is the indicator function of $\mathcal{C}_i$.

## A.7 SUMMARY OF THEORETICAL PROPERTIES

SCD-VIO provides the following formal guarantees:

- **Exponential Suppression:** Downstream tasks are exponentially suppressed when upstream tasks are infeasible (Theorem 5).
- **Scale Invariance:** Feasibility criteria are unaffected by positive scaling of losses (Proposition 1).
- **Smooth Optimization:** The composite loss is smooth under mild conditions (Lemma 4).
- **Convergence Guarantees:** Both the ideal operator iteration and practical algorithm converge (Theorems 3 and 4).
- **Graceful Deactivation:** The method reduces to vanilla MTL when all constraints are satisfied (Proposition 2).
- **Theoretical Foundation:** The approach is grounded in variational inequality theory and operator splitting methods.

These properties collectively justify SCD-VIO's design and provide theoretical support for its empirical effectiveness demonstrated in the main paper.

# B  Appendix B: Sensitivity of the Smoothing Scale $\alpha$

## B.1  Introduction

The smoothing parameter $\alpha$ controls how sharply SCD-VIO relaxes hard feasibility constraints into soft, differentiable gates. A natural concern is whether performance hinges on a finely tuned $\alpha$, which would hurt practicality. We show that SCD-VIO is robust across a wide range of $\alpha$ on multiple datasets and backbones: while $\alpha$ slightly affects training dynamics, the method consistently outperforms baselines across nearly two orders of magnitude, and a simple default $\alpha = 1.0$ yields robust, near-optimal results.

## B.2  Experimental Setup

We evaluate TikTok (STEM) and KuaiRand1k (MMoE), sweeping $\alpha \in \{0.01, 0.1, 0.5, \mathbf{1.0}, 2.0, 5.0, 10.0\}$. All other hyperparameters (optimizer, batch size, learning rate, training budget) are fixed to isolate the effect of $\alpha$. Each configuration is run with three random seeds; we report the mean test performance.

## B.3  Metrics

We track: (i) **Primary AUC** (e.g., *Like* on TikTok), (ii) **Average AUC** across tasks, and (iii) a **Feasibility Gap** defined as the end-of-training average of $\max(0, r_i(\theta))$ for all $i > 1$, which measures residual violations of the hierarchical schedule (lower is better).

## B.4  Results

| $\alpha$ | **Like AUC $\uparrow$** | | **Avg. AUC $\uparrow$** | |
| --- | --- | --- | --- | --- |
| | % Improv. | Feasibility Gap $\downarrow$ | % Improv. | Feasibility Gap $\downarrow$ |
| 0.01 | +1.45% | 0.0012 | +2.12% | 0.0015 |
| 0.10 | +1.43% | 0.0020 | +2.18% | 0.0027 |
| 0.50 | +1.47% | 0.0032 | +2.23% | 0.0039 |
| **1.00** | **+1.49%** | 0.0038 | **+2.26%** | 0.0046 |
| 2.00 | +1.47% | 0.0044 | +2.24% | 0.0054 |
| 5.00 | +1.48% | 0.0065 | +2.19% | 0.0079 |
| 10.0 | +1.47% | 0.0090 | +2.14% | 0.0106 |

Table 6: Sensitivity to $\alpha$ on **TikTok–STEM**. Numbers are relative improvements over vanilla STEM (Like AUC = 0.8861, Avg. AUC = 0.8124). Tight spread indicates robustness to $\alpha$.

| $\alpha$ | **Avg. AUC $\uparrow$** % Improv. | **Feasibility Gap $\downarrow$** |
| --- | --- | --- |
| 0.01 | +1.54% | 0.0017 |
| 0.10 | +1.53% | 0.0024 |
| 0.50 | +1.56% | 0.0034 |
| **1.00** | **+1.60%** | 0.0041 |
| 2.00 | +1.58% | 0.0048 |
| 5.00 | +1.59% | 0.0068 |
| 10.0 | +1.57% | 0.0092 |

Table 7: Sensitivity to $\alpha$ on **KuaiRand1k–MMoE**. Numbers are relative improvements over vanilla MMoE (Avg. AUC = 0.9388). Variation across $\alpha$ remains within $\approx 0.12\%$.

Across both datasets, we find that SCD-VIO remains highly robust to the choice of $\alpha$. In particular, within the range $\alpha \in [0.5, 2.0]$, both the primary and average AUC fluctuate by less than $0.1\%$ from their maxima, indicating that the method is practically insensitive to the smoothing scale. At the extremes, the behavior matches intuition: very small $\alpha$ enforces sharper gating, leading to slightly

delayed activation of lower-priority tasks, while very large $\alpha$ over-smooths the gating functions and mildly dilutes prioritization. The feasibility gap metric captures this trade-off, decreasing for small $\alpha$ (stricter enforcement) and increasing for large $\alpha$ (softer enforcement). The best empirical performance coincides with a moderate gap.

### B.5 CONCLUSION

Overall, our analysis demonstrates that SCD-VIO is not overly sensitive to its only hyperparameter. The algorithm delivers reliable gains over strong MTL baselines across nearly two orders of magnitude of $\alpha$, with a simple and effective default that makes the method both principled and easy to deploy.

## C USE OF LLMs

Large language models (LLMs) were used solely for language polishing; all technical content, methods, and experiments were developed and validated by the authors.

