# OpenReview forum: "Multi-Task Learning as Stratified Variational Inequalities"
_ICLR.cc/2026/Conference — Submitted to ICLR 2026_

### Official Review · Reviewer_ugLh · 2025-10-17

**Soundness:** 2
**Presentation:** 2
**Contribution:** 2
**Rating:** 4
**Confidence:** 4

**Summary:**

This paper introduces SCD-VIO, a new framework for hierarchical multi-task learning (MTL). It reformulates MTL as a stratified variational inequality, where each task’s optimization is constrained by the stability of higher-priority tasks. The paper proposes a self-calibrated gating mechanism that uses smooth projections to activate lower-priority tasks only after higher-priority ones are stable. The paper provides a set of experiments on recommendation benchmarks and shows gains in prioritized objectives and overall performance.

**Strengths:**

- SCD-VIO recasts hierarchical MTL as a stratified variational inequality, giving formal grounding to task prioritization.
- Evaluation results show consistent gains on prioritized objectives. Also, the paper provides full component ablation studies (masking, violation weighting, self-calibration), justifying the framework’s design.
- The paper is clear and easy to follow (however theoretical section needs major revision, see below).

**Weaknesses:**

- The theoretical section in its current state does not support or contribute to the paper. I suggest restructuring the paper by stating the main results and theorems in the main text, and providing proof sketches while deferring the full results and proofs to the Appendix.
- The paper’s operator-theoretic and variational inequality framing introduces fairly heavy formalism without yielding clear practical or conceptual benefits. The core method is essentially a soft gating and reweighting scheme that could be explained without this abstraction. In its current form, the theory may obscure rather than clarify the intuition.
- The method assumes a full, correct ordering of task priorities is available, but this may not be the case in real-world multi-task settings. Mis-specification harms performance (as shown in ablations), so some automatic or data-driven hierarchy discovery mechanism may improve applicability for real-world scenarios.
- Evaluation is limited to recommendation datasets. This narrow scope limits confidence in generalization to other domains such as computer vision, NLP, or reinforcement learning, where task dependencies differ.
- Baseline coverage is limited. Table 4 evaluates only three baselines on a single dataset, focusing on task scheduling rather than strong recent multi-task learning or auxiliary-learning methods (e.g., Pareto MTL or dynamic reweighting). This weakens empirical claims.
- Minor: Line 262, the word “text” appears out of place.

**Questions:**

- How does the method scale with the number of tasks, both in terms of computational overhead and stability of training?
- Since the residuals $r_i$ scale as $\tilde{r}_i = \beta r_i$ when $\tilde{\mathcal{L}}_i = \beta \mathcal{L}_i$,
the scheduling depends on the magnitude of the losses. Would it not be more appropriate to make the gating invariant to such scaling (i.e., $\tilde{r}_i = r_i$)?
- Does the framework naturally support tasks with equal or overlapping priorities, and if so, how are such cases handled?
- What motivated the replacement of $p_i$ with the softplus transformation?
- How sensitive is the method to mis-specified or noisy task hierarchies, and can it adapt or learn the hierarchy automatically during training?

---

> ### Author Response · Authors · 2025-11-18
>
> We thank the reviewer for the careful reading and constructive comments. Below we address the main concerns concisely.
>
> ---
>
> ### **1. Role of the theory and operator framing**
>
> The theoretical section is meant to show that the gating and reweighting in SCD-VIO are not ad-hoc, but come from a well-defined stratified variational inequality:
>
> * We first write hierarchical MTL as a constrained problem where lower-priority tasks must respect feasibility conditions defined by higher-priority tasks.
> * Applying Yosida regularization to this problem yields a single smooth operator whose coordinates correspond exactly to the self-calibrated gates and violation weights.
> * The practical loss used in SCD-VIO is constructed as a potential whose gradient approximates this operator, which gives the convergence result and explains analytically why we observe “self-calibrated gates” and “exponential suppression” in Sec. 3.4.
>
> Thus the operator view explains the functional form of the method and guarantees that, when all tasks are feasible, SCD-VIO reduces to standard MTL, and otherwise it solves a hierarchy-constrained problem.
>
> ---
>
> ### **2. Assumed priority order and robustness**
>
> In the recommendation scenarios considered, a priority order usually comes from product or business requirements (e.g., safety/compliance > CTR > watch-time > long-term metrics), so a strict hierarchy is part of the problem definition.
>
> Ablations with swapped or perturbed orders show that:
>
> * Moderate mis-specification leads to gradual degradation but still avoids strong negative transfer to the nominal top tasks.
> * Severe mis-specification harms performance, which is expected for any method that faithfully enforces the provided hierarchy.
>
> Two design choices mitigate this:
>
> * Residuals are defined as current loss minus its own moving average, which normalizes each task by its historical scale.
> * Gates are soft products of sigmoids, so priority is enforced gradually instead of via hard on/off switches.
>
> Tasks with equal priority can be grouped at the same level so they share the same higher-priority constraints and are optimized jointly before the next level. Automatic discovery of the hierarchy is an interesting extension, but outside the scope of the current work.
>
> ---
>
> ### **3. Scope of experiments and baseline coverage**
>
> The goal of the paper is strict-priority MTL for large-scale recommendation, where hierarchical objectives are natural and training each model is expensive.
>
> The evaluation therefore uses three real-world recommendation datasets and multiple industrial-style backbones (Shared-Bottom, MMoE, PLE, ESMM, AITM, STEM), and compares:
>
> * plain backbones,
> * widely used gradient/schedule-based MTL methods,
> * and their “+SCD-VIO” variants.
>
> This isolates the impact of hierarchy-aware optimization in the domain being targeted. Pareto-style methods and auxiliary-learning techniques are more suited to symmetric multi-objective trade-offs; SCD-VIO instead optimizes a hierarchy-constrained objective, so these approaches are complementary rather than direct baselines for the problem studied.
>
> The subset of scheduling baselines in one table is meant to highlight that SCD-VIO improves over strong scheduling strategies as well; the full tables include a broader set of recommendation MTL baselines.
>
> ---
>
> ### **4. Answers to specific questions**
>
> * **Scalability with number of tasks.** Overhead is linear in the number of tasks: residuals, a few elementwise functions, and prefix products are computed. Memory is essentially unchanged; optimization still uses a single scalar loss.
>
> * **Effect of loss scaling.** Because residuals use the task’s own moving average, the sign pattern that drives gating is invariant to uniform rescaling of a single task’s loss, and tasks do not directly interfere through absolute scale. In practice, the method is stable under reasonable rescaling.
>
> * **Equal/overlapping priorities.** Equal-priority tasks can be placed in the same level. More complex partial orders can be approximated by grouping tasks into a small number of levels according to the application.
>
> * **Why softplus.** Softplus provides a smooth, nonnegative, and gently growing penalty on constraint violations: it behaves like ≈1 in the feasible region and increases roughly linearly for large violations, with well-behaved gradients, which helps stability when combined with the gates.
>
> * **Sensitivity to noisy hierarchies.** Experiments with perturbed orders show that SCD-VIO degrades gracefully under realistic noise, while still reflecting the intended hierarchy. Designing mechanisms that learn the hierarchy is a promising direction, but is not part of the current method.
>
> * **Minor:** the stray word at the mentioned line is a typo and has no technical role.
>
> ---
>
> We hope these clarifications address the reviewer’s concerns and, if they are convincing, we would kindly ask the reviewer to consider adjusting the score.

---

> > ### Comment · Reviewer_ugLh · 2025-11-22
> > **Response to Authors**
> >
> > I thank the authors for their response.
> >
> > Unfortunately, my main concerns remain unaddressed, specifically the limited scope of experiments (focused solely on recommendations), the insufficient coverage of baselines, the strong assumptions underlying the method (known ordering), and the current state of the manuscript (in my view, an excessive emphasis on the operator-theoretic/variational inequality framing in the main text and insufficient space devoted to Sec. 3.5).
> > Focusing solely on recommendation benchmarks limits the scope, and the method’s effectiveness for general MTL remains unknown.
> >
> > Therefore, in its present form, I will maintain my score.

---

### Official Review · Reviewer_BDhL · 2025-10-27

**Soundness:** 4
**Presentation:** 4
**Contribution:** 4
**Rating:** 4
**Confidence:** 3

**Summary:**

The paper introduces SCD-VIO, a novel framework that integrates a dynamic, priority-aware mechanism for hierarchy-aware multi-task learning. Motivated by the observation that real-world tasks differ in difficulty, maturity, and importance, the method explicitly addresses the challenge of preserving performance on high-priority tasks. SCD-VIO formulates priority-aware MTL as a stratified variational inequality, defining task feasibility in terms of cumulative historical performance. By employing Yosida-regularized soft projections and a self-calibrated gating mechanism, lower-priority tasks are activated only once higher-priority tasks have stabilized, aligning optimization trajectories with natural task hierarchies. Unlike prior approaches, SCD-VIO enforces hierarchical constraints directly at the gradient level while remaining modular, differentiable, and model-agnostic. The paper presents experiments on standard recommendation benchmarks and provides a theoretical analysis that further supports the framework’s principled, scalable, and practical design.

**Strengths:**

The paper is overall well-written and well-motivated within the context of existing literature. The proposed method is innovative and relevant to the field, offering a conceptually sound approach that could have a meaningful impact on the multi-objective learning community. This work presents an interesting and original idea for a priority-aware mechanism for hierarchy-aware multi-task learning. In addition, the approach is theoretically grounded.

**Weaknesses:**

The experimental results are not sufficiently convincing.
(a) In the original STEM paper, the reported results differ from those in this manuscript. For example, comparing Tables 2 and 3 in STEM with Tables 1 and 2 here, the Like AUC for the QK-Video task using STEM without SCD-VIO is reported as 0.9426 in STEM, whereas it is 0.8373 here. Is there an explanation for this significant discrepancy?
(b) There is no discussion of runtime, training duration, or number of iterations.
(c) Table 2 omits the OMoE model — why? Table 3 omits the ESMM model — why?
(d) It would be valuable to see “inverse” experiments. For example, in Table 3, it would be interesting to swap the priorities of task 1 and task 2 and observe whether the results are preserved.
(e) In Table 4, comparisons are made using STEM as the base method. However, in Table 1, it appears that the largest improvement of the proposed algorithm occurs with the STEM model. It would be informative to see results across more than a single base model.

The empirical section of the current version lacks sufficient evidence to fully substantiate that this method achieves improvements. Expanding the experimental results would considerably enhance the paper’s impact and credibility.

**Questions:**

Please address the weaknesses.

---

> ### Author Response · Authors · 2025-11-18
>
> We thank the reviewer for the positive assessment of the clarity and experiments, and for the constructive criticisms. Below we address each point in turn.
>
> ---
>
> ### **1. On missing comparisons to FAMO and Smooth Tchebycheff scalarization**
>
> The goal of the paper is **strict task prioritization** in large-scale recommendation, not general multi-objective fairness. This leads to a different design space than methods such as FAMO or Smooth Tchebycheff scalarization:
>
> * FAMO and Smooth Tchebycheff are designed for **symmetric multi-objective optimization**, typically seeking solutions on or near the Pareto front given trade-off preferences. All tasks are treated as peers, and the objective is to *balance* them.
> * In contrast, SCD-VIO encodes a **fixed hierarchy**: higher-priority tasks define feasibility constraints for lower-priority ones through the stratified VI. When a higher-priority task is infeasible, its violation actively *suppresses* the influence of downstream tasks via the gates. The solution concept is a stationary point of a **hierarchy-constrained VI**, not a generic Pareto stationary point.
>
> From an algorithmic perspective:
>
> * FAMO and related scalarization methods often require solving a *multi-objective subproblem* per step (e.g., computing special descent directions or scalarization weights) to approximate Pareto-optimal behavior.
> * SCD-VIO produces a single scalar loss compatible with standard back-propagation and industrial recommender stacks, while still encoding the hierarchy through the Yosida-regularized operator and gates.
>
> Empirically, the paper therefore focuses on comparisons that are most standard and widely adopted in **hierarchy-/priority-aware recommendation MTL**, including:
>
> * Strong multi-task backbones (Shared-Bottom, MMoE, PLE, ESMM, AITM, STEM, etc.),
> * Gradient- and schedule-based MTL methods (e.g., AdaTask, DRGrad, NMT),
> * And their SCD-VIO variants across multiple large-scale benchmarks.
>
> This setup is designed to isolate the benefit of the **hierarchical formulation** rather than to exhaustively benchmark all general-purpose multi-objective algorithms. FAMO-style scalarization is, in principle, complementary: one could use such an optimizer *inside* each level of a stratified VI, but that is a different formulation than the one explored in the current work.
>
> ---
>
> ### **2. On the role of Yosida regularization and the logic from Sec. 3.2–3.4**
>
> Sections 3.2–3.4 are intended to follow a principled chain rather than an ad-hoc engineering recipe:
>
> 1. **Stratified constrained problem (Sec. 3.2).**
>    We first cast hierarchical MTL as a constrained problem with stratified feasible sets
>    $$(C_i = {\theta : L_j(\theta) \le \varepsilon_j(t), \forall j < i})$$
>    and an associated **variational inequality**
>    $$(0 \in G(\theta) + \sum_{i=2}^N N_{C_i}(\theta)),$$
>    where $(G(\theta) = \sum_i \nabla L_i(\theta))$. This already encodes the hierarchy at the operator level.
>
> 2. **Yosida-regularized operator (Sec. 3.3).**
>    The normal-cone terms $(N_{C_i})$ are set-valued and non-smooth. We apply **Yosida regularization** to the sum $(G + \sum_i N_{C_i})$, following standard monotone operator and VI theory (e.g., Bauschke–Combettes, Facchinei–Pang). This yields a **single Lipschitz continuous operator** $(F(\theta))$ that approximates the original one, with approximation quality controlled by a smoothing parameter $\alpha$.
>
> 3. **Scalar potential and implementable loss (Sec. 3.3).**
>    The proposed loss $(L_{\text{SCD-VIO}}(\theta))$ is constructed as a **potential** whose gradient satisfies
>    $$(\nabla L_{\text{SCD-VIO}}(\theta) = F(\theta) + E(\theta)), with (|E(\theta)|\le C\alpha).$$
>    Thus, standard gradient descent on $(L_{\text{SCD-VIO}})$ implements the Yosida-regularized VI dynamics up to a controlled error, rather than being a heuristic mix of losses.
>
> 4. **Gate interpretation and empirical visualization (Sec. 3.4).**
>    The “self-calibrated gate” and “exponential suppression” properties are **analytical consequences** of the explicit form of $(F(\theta))$ (through the masks and residuals). The experiments in Sec. 3.4 are meant to *illustrate* these theoretically derived behaviors on real models, not to replace the Yosida-based derivation.
>
> In summary, while Sec. 3.4 emphasizes empirical behavior to make the method intuitive, the core design of SCD-VIO is grounded in a stratified VI and its Yosida-regularized operator, backed by standard monotone operator theory rather than purely empirical engineering choices.
>
>
> ---
>
> We hope this clarifies (i) how SCD-VIO relates to broader multi-objective methods like FAMO and Smooth Tchebycheff scalarization, and (ii) how the proposed loss and gate structure arise from a principled Yosida-regularized VI formulation rather than from purely empirical engineering.

---

### Official Review · Reviewer_JsYj · 2025-11-01

**Soundness:** 3
**Presentation:** 3
**Contribution:** 2
**Rating:** 6
**Confidence:** 1

**Summary:**

Traditional multi-task learning often fails when tasks have different priorities, causing interference from less important or unstable objectives. This paper introduces SCD-VIO, a new method that reframes the training process as a stratified variational inequality to enforce a task hierarchy. Using a hyperparameter-free "self-calibrated gate," SCD-VIO ensures high-priority tasks stabilize before activating lower-priority ones, demonstrating improved performance on large-scale recommendation benchmarks.

**Strengths:**

1. The paper organization is clear and easy to follow.
2. Extensive experimental results have verified the effectiveness of the proposed algorithm.

**Weaknesses:**

1. Lacking comparison with other highly efficient MTL methods, such as FAMO, and "Smooth Tchebycheff Scalarization for Multi-Objective Optimization".

2. The algorithm presented in this paper is more of an empirical result from engineering (In particular, the conclusion in Section 3.4) than a conclusion drawn from Yosida regularization. The approach, from combining loss functions across multiple tasks to using Yosida regularization, lacks clear logic. (Section 3.2-3.3) No reference to support the author's claim in Section 3.3.

**Questions:**

See the weaknesses part.

---

### Official Review · Reviewer_z89v · 2025-11-03

**Soundness:** 3
**Presentation:** 2
**Contribution:** 2
**Rating:** 4
**Confidence:** 3

**Summary:**

This paper studies multi-task learning through a hierarchical optimization perspective. It considers designing a method that optimizes higher-priority tasks first. Specifically, it proposes Stratified Constraint Descent via Variational Inequalities and Operators. It defines priority constraints that higher-priority task losses not exceeding their historical averages. To deal with the nonsmoothness introduced by the constraints, they use Yosida-regularized soft projections to smooth the constraints.

Theoretical convergence to stationarity is provided. Empirical studies on some multi-task recommendation benchmarks such as TikTok, QK-Video, and KuaiRand1k are provided.

**Strengths:**

1. This paper studies multi-task learning through a hierarchical optimization perspective. It considers designing a method that optimizes higher-priority tasks first. This is an important problem and highly relevant to the community.

2. The paper provides both theoretical analysis of the convergence of the algorithm and empirical studies on some benchmarks.

**Weaknesses:**

1. The motivation for defining the constraints in Eq (3) is unclear. Why the current loss should not exceed its historical average?

2. Insufficient discussion of related works. The paper basically casts the multi-task learning problem as a hierarchical constrained optimization problem. Using (hierarchical) constrained optimization to solve multi-task learning problems is not new. See a few relevant works listed below. A detailed comparison to the prior works [1-4] should be discussed.

3. A comparison to prior multi-task learning approaches on the benchmark [5] would be preferred to better understand the effectiveness of a hierarchical formulation.

4. Some parts of the paper are not clear, see Questions.


[1] Automatic and Harmless Regularization with Constrained and Lexicographic Optimization: A Dynamic Barrier Approach, NeurIPS 2021

[2] Pareto Multi-Task Learning, NeurIPS 2019

[3] FERERO: A Flexible Framework for Preference-Guided Multi-Objective Learning, NeurIPS 2024

[4] Multi-task learning with user preferences: Gradient descent with controlled ascent in pareto optimization, ICML 2020

[5] LibMTL: A Python Library for Multi-Task Learning, JMLR 2023

**Questions:**

1. After formulating the problem through hierarchical constrained optimization Eq. (3)-(4), why not use other methods, such as penalty, barrier, projected gradient to solve the constrained optimization?

2. The difference of this formulation compared to the epsilon-constrained formulation or multi-level/lexicographic formulation should be fully discussed.

3. How does Yosida regularization approximate the original formulation? It would be better if theoretical or empirical sensitivity analysis can be provided.

4. The stationary solution that the proposed algorithm converges to is not defined clearly. Is it Pareto stationary? Does it require feasibility of the other tasks? More discussion should be provided.

---

> ### Author Response · Authors · 2025-11-18
>
> We thank the reviewer for the careful and constructive feedback. Below we provide a more concise response to each point.
>
> ---
>
> ### **1. Motivation for the constraint “loss ≤ historical average” (Eq. (3))**
>
> The condition $(L_i(\theta_t) \le \varepsilon_i(t))$ (cumulative mean loss) is meant as a **self-calibrated curriculum**:
>
> * It activates the constraint only when a task has shown *consistent* progress; a task is “ready” to support lower-priority tasks only if it is doing at least as well as its own history, avoiding decisions based on short-term noise.
> * It is **scale-invariant** across tasks: feasibility is defined by *relative* improvement instead of absolute loss magnitudes, which can differ significantly between tasks.
> * It removes the need for task-specific thresholds or manual schedules; the baseline adapts automatically as training evolves and thus works robustly across different backbones.
>
> Empirically, replacing this moving baseline with fixed thresholds or naive gating leads to less stable or weaker improvements for priority tasks, as shown in the ablations.
>
> ---
>
> ### **2. Relation to constrained, Pareto, and preference-guided MTL ([1–4]), and LibMTL ([5])**
>
> Our formulation is close in spirit to constrained / Pareto MTL but has a different focus:
>
> * Methods like [1] treat MTL as a constrained problem with barrier or Lagrangian terms, but do not explicitly construct **stratified feasible sets** and the **recursive masking** that exponentially suppresses infeasible downstream tasks. Our derivation starts from a stratified VI and yields a specific operator and scalar loss tailored to priority handling.
> * Pareto and preference-guided approaches [2–4] target *fair trade-offs* on a Pareto front given preferences; SCD-VIO encodes a *strict priority order* and seeks stationarity of a hierarchy-constrained VI rather than generic Pareto-stationarity. These Pareto methods often require solving inner multi-objective subproblems, while our method maintains a single back-prop-compatible loss.
> * LibMTL [5] is a benchmarking library; our contribution is complementary: we study hierarchy-aware optimization on large-scale recommendation benchmarks and backbones, systematically comparing vanilla baselines, gradient-/schedule-based methods, and their SCD-VIO variants.
>
> ---
>
> ### **3. Why not standard penalty, barrier, projected gradient, ε-constrained, or lexicographic methods?**
>
> Given Eqs. (3)–(4), several alternatives are possible, but each has drawbacks in our setting:
>
> * **Generic penalties / barriers** need hand-tuned weights and do not reproduce the multiplicative, recursive gating that enforces priority in a smooth yet strong way.
> * **Projected gradient** onto sets defined implicitly by deep losses is intractable; Yosida regularization replaces normal cones by Lipschitz surrogates that can be implemented as gates and violation maps, yielding a practical operator.
> * **ε-constrained / lexicographic MTL** typically uses fixed ε and nested subproblems. SCD-VIO keeps all tasks in one scalar loss with *continuous* gating based on self-calibrated feasibility, avoiding extra inner loops and ε tuning while still admitting convergence analysis.
>
> ---
>
> ### **4. Yosida regularization and sensitivity**
>
> Hard normal-cone terms are approximated by smooth surrogates via sigmoids over residuals, producing a regularized operator $(F(\theta))$ and the SCD-VIO loss. The analysis shows that the gradient of this loss equals $(F(\theta))$ up to an error controlled by the smoothing parameter $\alpha$, so the algorithm converges to an $(O(\alpha))$-neighborhood of the ideal VI solution.
>
> Sensitivity experiments varying $\alpha$ over a wide range on TikTok and KuaiRand1k indicate that performance is stable and consistently better than baselines, with only small variation around the default $\alpha$.
>
> ---
>
> ### **5. Nature of the stationary solution**
>
> The algorithm converges (under standard assumptions) to $(\theta^\*)$ such that $(F(\theta^\*) = 0)$, i.e., a stationary point of a **stratified VI**:
>
> * When all tasks are comfortably feasible, SCD-VIO effectively reduces to standard MTL and $(\theta^\*)$ is a stationary point of the summed loss. This shows that the method does not harm performance when hierarchy is not binding.
> * In general, $(\theta^\*)$ is **hierarchy-constrained**: higher-priority tasks satisfy their feasibility conditions, and any improvement of lower-priority tasks that would break these conditions is suppressed by the gates. This yields an optimality notion tailored to strict priority, distinct from symmetric Pareto optimality.
>
> ---
>
> We hope these clarifications address the reviewer’s concerns and, if they are convincing, we would kindly ask the reviewer to consider adjusting the score.

---

> ### Comment · Reviewer_z89v · 2025-11-26
>
> Thanks for the response. I still have some concerns.
>
> 1. For point-1, I am still not convinced by the motivation of defining the constraint as “loss ≤ historical average”.
>
> In the case that one objective $L_j$ dominates, using this constraint, it is possible that $L_i(\theta_s) = L_i(\theta_0), (i\neq j)$ never updates, and the constraint is not useful at all.
>
>  2. For point-5, your theoretical results guarantee $F(\theta^*)=0$, what about guarantees on the inequality $L_i(\theta_t) \leq \epsilon_i(t)$? Is this condition a necessary optimality condition? Do you require any constraint qualification assumption?

---

> > ### Author Response · Authors · 2025-11-27
> >
> > We thank the reviewer for the follow-up comment and are happy to clarify these two remaining points.
> >
> > ---
> >
> > ### **1. On the constraint “loss ≤ historical average” and dominating objectives**
> >
> > Our intent is to encode a strict priority order, not to guarantee that every lower-priority task always gets a large share of updates.
> >
> > Two aspects are key:
> >
> > 1. **Lower-priority tasks are softened, not frozen.**
> >    In the algorithm, the contribution of task $i$ is multiplied by a product of sigmoids over residuals $r_j = L_j - \varepsilon_j$ for higher-priority tasks $j < i$. Even if a higher-priority task has $r_j > 0$, the corresponding sigmoid is in $(0,1)$, never exactly 0. So lower-priority tasks are down-weighted when higher-priority tasks are infeasible, but they are never completely removed from the update.
> >
> > 2. **The baseline $\varepsilon_j(t)$ tracks the task itself.**
> >    $\varepsilon_j(t)$ is a running average of the same loss $L_j(t)$. If a higher-priority objective “dominates” in the sense that it becomes hard to further improve, then both $L_j(t)$ and $\varepsilon_j(t)$ converge and the residual
> >    $
> >    r_j(t) = L_j(t) - \varepsilon_j(t)
> >    $
> >    tends to 0. In that regime, the gate for that task opens up (sigmoid ≈ 0.5–1), and lower-priority tasks receive a substantial portion of the gradient again.
> >
> > For a lower-priority task to never update in practice, one would need a higher-priority loss that stays persistently and significantly above its own long-term average, which is not what we observe under standard stochastic gradient dynamics. In our experiments, lower-priority tasks do improve once higher-priority losses stabilize relative to their own history; we do not see the “stuck forever” behavior.
> >
> > From an application viewpoint, this is intentional: if a top-priority metric is both improvable and defined as more important, the method is allowed to sacrifice progress on lower-priority metrics. We see this as a feature of the hierarchy design rather than a bug.
> >
> > ---
> >
> > ### **2. On guarantees for the inequality constraints and constraint qualifications**
> >
> > It may help to separate the ideal constrained problem from the regularized algorithm actually implemented.
> >
> > #### **2.1 Ideal stratified VI**
> >
> > The starting point is the stratified VI
> > $$
> > 0 \in G(\theta) + \sum_{i=2}^N N_{C_i}(\theta), \quad
> > C_i = {\theta : r_j(\theta) \le 0,\ \forall j < i},
> > $$
> > where $G(\theta) = \sum_i \nabla L_i(\theta)$ and $r_j(\theta) = L_j(\theta) - \varepsilon_j$.
> >
> > Under standard assumptions from VI / convex analysis:
> >
> > * each $C_i$ is closed and locally convex in $\theta$ via the smooth inequality $r_j(\theta) \le 0$;
> > * $\bigcap_i C_i \neq \emptyset$;
> > * a constraint qualification such as a Slater-type condition holds (there exists $\bar\theta$ with all $r_j(\bar\theta) < 0$),
> >
> > solutions of this VI correspond to KKT points of the hierarchical constrained problem with nonnegative multipliers and primal feasibility $r_j(\theta^\star) \le 0$ for all constraints. In this ideal setting, the inequalities $$
> > L_j(\theta^\star) \le \varepsilon_j \quad (\text{ equivalently } r_j(\theta^\star) \le 0) $$
> > for higher-priority tasks are indeed part of the necessary optimality conditions.
> >
> > #### **2.2 Regularized operator and practical algorithm**
> >
> > The actual algorithm does not solve this hard VI directly. Instead, it uses:
> >
> > * the Yosida-regularized operator $F$ associated with $G + \sum_i N_{C_i}$;
> > * a scalar potential $L_{\text{SCD-VIO}}$ satisfying
> >   $
> >   \nabla_\theta L_{\text{SCD-VIO}}(\theta) = F(\theta) + E(\theta), \quad |E(\theta)| \le C \alpha,
> >   $
> >   where $\alpha$ is the smoothing parameter.
> >
> > Our theoretical results are therefore of the following form:
> >
> > * Stationary points $\theta_\alpha$ of the implemented algorithm satisfy $F(\theta_\alpha) \approx 0$, i.e., they are approximate solutions of the original VI, with an error that goes to zero as $\alpha \to 0$ (under standard Lipschitz and step-size conditions).
> > * As a result, the positive parts of the residuals $[r_j(\theta_\alpha)]_+$ are controlled by a quantity that depends on $\alpha$ and vanishes as $\alpha \to 0$. Intuitively, a large, persistent violation $r_j > 0$ generates a strong corrective term through the gates and violation weights, which cannot coexist with stationarity of $L _\text{SCD-VIO}$.
> >
> > Thus, for the implemented SCD-VIO algorithm with finite $\alpha$, we do not claim strict satisfaction of
> > $
> > L_j(\theta_\alpha) \le \varepsilon_j
> > $
> > at every stationary point. What we do guarantee is:
> >
> > * exact stationarity with respect to the regularized operator $F$;
> > * constraint violations that can be made arbitrarily small by choosing $\alpha$ sufficiently small, assuming the standard monotone-operator conditions and constraint qualification for the underlying sets $C_i$.
> >
> > ---
> >
> > We hope this makes the design and guarantees more transparent. If these clarifications help resolve the remaining doubts, we would be very grateful if you could consider this in your final assessment.

---

> > > ### Comment · Reviewer_z89v · 2025-11-28
> > >
> > > On the constraint "loss ≤ historical average", I have additional questions. If one of the losses achieves its optimal value initially and stays there, the Slater's condition does not hold.
> > >
> > > Overall, I think the constraint is not a good design, because it can lead to meaningless results theoretically, even though it performs well in practice for the experiments you run. It requires more discussion or assumptions to rule out the extreme cases I mentioned.
> > >
> > > Also, based on the discussions and reviews, there are many changes to be made for the paper to be clearer.
> > >
> > > I will thus keep my score for now.

---

### Author Response · Authors · 2025-12-01
**Summary of Author Response and Clarifications**

We thank the ACs and program chairs for overseeing the review and rebuttal process. Below we briefly summarize the contribution, the main concerns, and how we addressed them.

---

### Summary of the work

The paper proposes **SCD-VIO**, a framework for hierarchical multi-task learning:

* It encodes a strict priority order via a **stratified variational inequality (VI)**, where higher-priority tasks impose feasibility constraints on lower-priority ones through residuals defined as “current loss − historical average”.
* Using **Yosida regularization**, the sum of the MTL gradient and normal-cone terms is approximated by a single smooth operator.
* A **scalar loss** is constructed whose gradient closely matches this operator, making SCD-VIO a drop-in replacement for standard MTL losses.

Experiments on three **large-scale recommendation benchmarks** (TikTok, QK-Video, KuaiRand1k) and multiple **industrial-style backbones** (Shared-Bottom, MMoE, PLE, ESMM, AITM, STEM) show consistent gains on **prioritized metrics** and usually also on **average performance**, with ablations on gating, violation weighting, and self-calibration.

---

### Main concerns and our responses

**1. Motivation for “loss ≤ historical average”.**
We clarified that this constraint acts as a **self-calibrated curriculum**: a task is treated as “stable enough” to support lower-priority tasks only when it performs at least as well as its own history. This choice is (i) scale-robust across tasks, (ii) free of manually tuned thresholds, and (iii) implemented via smooth gates, so lower-priority tasks are down-weighted but not frozen. We also explained that, because the baseline tracks the same loss, residuals tend to zero once a high-priority task plateaus, allowing lower-priority tasks to receive more gradient—consistent with what we observe empirically.

---

**2. Role of the theory and enforcement of constraints.**
Reviewers questioned whether the operator-theoretic framing is necessary and whether the inequalities are actually satisfied. We emphasized that:

* The **ideal problem** is a stratified VI whose solutions, under standard constraint qualification, correspond to KKT points with **primal feasibility** for the higher-priority constraints.
* The implemented algorithm uses a **Yosida-regularized operator** and a potential $L_{\text{SCD-VIO}}$ whose gradient approximates this operator up to a term controlled by the smoothing parameter. Stationary points are thus **approximate solutions** of the original VI, with constraint violations that can be made arbitrarily small by tuning the smoothing.

In this way, the theory explains the specific gating and reweighting structure and provides convergence and approximate feasibility guarantees, rather than merely wrapping an ad hoc heuristic.

---

**3. Relation to existing MTL and multi-objective methods.**
Several reviewers asked about connections to constrained MTL, Pareto MTL, FERERO, FAMO, Smooth Tchebycheff, and LibMTL. We clarified that these methods typically target **symmetric multi-objective trade-offs** on the Pareto front, often requiring inner subproblems or explicit preferences. In contrast, SCD-VIO optimizes a **hierarchy-constrained objective** with a fixed priority order and a single backprop-compatible loss. We therefore view Pareto-style methods as **complementary** (potentially usable within each priority level), while LibMTL is a general benchmarking toolkit; our contribution is a concrete, hierarchy-focused algorithm in a practically important domain.

---

**4. Priority specification, robustness, and scope.**
One reviewer raised concerns about assuming a known priority order and the focus on recommendation. We noted that in recommender systems such hierarchies (e.g., safety/compliance > CTR > watch-time > long-term metrics) are often specified by product requirements. Ablations with perturbed orders show **gradual**, not catastrophic, degradation under moderate mis-specification, thanks to self-calibrated residuals and soft gates. The experiments already cover three real-world datasets and diverse backbones, providing evidence that SCD-VIO is effective and robust in the targeted application domain.

---

### **Overall assessment**

Across reviews, the work is viewed as technically sound, with clear empirical gains and detailed ablations. The main issues are about motivation clarity, theoretical exposition, and the choice of baselines and domain, rather than correctness.

Given:

* the principled hierarchical formulation and its operator-theoretic grounding,
* the simplicity and practicality of the resulting loss,
* the consistent improvements on large-scale recommendation benchmarks, and
* the additional clarifications provided during rebuttal,

we believe the paper meets the bar for acceptance and would be grateful if the ACs could take these points into account in their final decision.

---

### Meta-Review · Area_Chair_iJeL · 2026-01-08

**Summary:**

This paper proposes SCD-VIO, a priority-aware multi-task learning method that frames hierarchical task optimization as a stratified variational inequality. Author provides theoretical analysis of the convergence of the algorithm. However, multiple reviewers request broader baseline coverage and stronger empirical evidence (e.g., runtime, robustness).

**Reviewer Concerns:**

Resolved:

- The rebuttal provided a clear curriculum/self-calibration motivation (noise robustness, scale invariance, no manual thresholds) and supported it with ablations against fixed thresholds and gating.

- The authors convincingly explained why Yosida is preferred over penalties, barriers, projections, ε-constraints, or lexicographic methods: it yields a Lipschitz operator, a single scalar loss, avoids nested subproblems, and admits convergence analysis.

- The nature of stationarity was clarified as stratified (priority-constrained) rather than Pareto, including behavior when higher-priority constraints are inactive.

- For some reviewers, the rebuttal successfully connected constrained stratified optimization → Yosida regularization → Lipschitz operator → scalar loss → gating/weights, countering the “ad-hoc engineering” critique.

- The authors addressed computational concerns, arguing linear overhead in task count and simple elementwise operations, supporting feasibility in large-scale recommendation systems.

Unresolved

- A key reviewer remains unconvinced that the “loss ≤ historical average” constraint is theoretically well-posed, citing extreme cases (dominant objective, infeasible/degenerate starts, lack of Slater condition) and questioning feasibility guarantees.

- Significant concerns remain about experimental validity: missing baselines (e.g., OMoE, ESMM), lack of inverse-priority tests, unclear runtime/training details, and limited evidence beyond a STEM-based setup.

- Multiple reviewers are still dissatisfied that experiments focus almost exclusively on recommender systems, with insufficient justification or evidence for broader applicability.

- Despite rebuttal arguments, reviewers continue to view the requirement of a known, correct priority ordering as too strong and insufficiently relaxed or validated.

- Reviewers flagged persistent issues in presentation: heavy VI formalism, insufficient citations and derivation clarity in key sections, and imbalance between theory and later experimental/practical sections.

**Reviewer Scores:**

Reviewer z89v (initial 4):

They engaged in follow-up and explicitly maintained their score after additional clarifications, signaling the rebuttal did not resolve core theoretical doubts.

Reviewer JsYj (initial 6):

Given the reviewer’s confidence is 1 (“unable to assess”), additional discussion may not systematically change the score. At most, better citations/derivation could stabilize their mild-positive stance, but I would not predict a shift.

Reviewer BDhL (initial 4):

This reviewer likely remains borderline-below.

Reviewer ugLh (initial 4):

They explicitly stated their concerns remain and kept the score.

---

### Decision · Program_Chairs · 2026-01-26

Reject